# Ubiquitin, Autophagy and Neurodegenerative Diseases

**DOI:** 10.3390/cells9092022

**Published:** 2020-09-02

**Authors:** Yoshihisa Watanabe, Katsutoshi Taguchi, Masaki Tanaka

**Affiliations:** 1Department of Basic Geriatrics, Graduate School of Medical Science, Kyoto Prefectural University of Medicine, Kyoto 602-8566, Japan; 2Department of Anatomy and Neurobiology, Graduate School of Medical Science, Kyoto Prefectural University of Medicine, Kyoto 602-8566, Japan; ktaguchi@koto.kpu-m.ac.jp

**Keywords:** ubiquitin, autophagy, neurodegenerative diseases, ubiquitin–proteasome system, autophagy–lysosome pathway

## Abstract

Ubiquitin signals play various roles in proteolytic and non-proteolytic functions. Ubiquitin signals are recognized as targets of the ubiquitin–proteasome system and the autophagy–lysosome pathway. In autophagy, ubiquitin signals are required for selective incorporation of cargoes, such as proteins, organelles, and microbial invaders, into autophagosomes. Autophagy receptors possessing an LC3-binding domain and a ubiquitin binding domain are involved in this process. Autophagy activity can decline as a result of genetic variation, aging, or lifestyle, resulting in the onset of various neurodegenerative diseases. This review summarizes the selective autophagy of neurodegenerative disease-associated protein aggregates via autophagy receptors and discusses its therapeutic application for neurodegenerative diseases.

## 1. Introduction

Many neurodegenerative diseases, such as Alzheimer’s disease (AD), Parkinson’s disease (PD), amyotrophic lateral sclerosis (ALS), and Huntington’s disease (HD), involve accumulation of harmful and aggregation-prone proteins. These aggregated proteins are known to be ubiquitinated in many neurodegenerative diseases. Although harmful proteins are immediately degraded by proteolytic systems in healthy individuals, any perturbation of these systems caused by genetic variation, aging, or lifestyle results in accumulation of harmful protein aggregates and the onset of various diseases including neurodegenerative diseases. Ubiquitination is the most important targeting signal for proteolytic systems [1]. Indeed, pathological analyses show that most of the protein inclusions and aggregates in the brains of neurodegenerative disease cases are positive for ubiquitin [2]. Recent advances in mass spectrometry technology have contributed to the characterization of ubiquitin chains and the decoding of ubiquitin signals. Ubiquitin signals are categorized as mono-ubiquitin, homotypic poly-ubiquitin, and heterotypic poly-ubiquitin [3]. Homotypic poly-ubiquitin chains are generated by conjugation of two or more ubiquitin molecules via their seven lysine residues (Lys-6, Lys-11, Lys-27, Lys-29, Lys-33, Lys-48, and Lys-63) or the initiation methionine residue (Met-1), whereas heterotypic poly-ubiquitin chains are formed by linkages of two or more different Lys residues [3]. These ubiquitin signals have roles in proteolytic functions and non-proteolytic functions, such as transcription regulation, membrane trafficking, DNA repair, and cell signaling [4]. Mutations in several autophagy related proteins, such as Parkin, PINK1, p62, and OPTN, are linked to neurodegenerative diseases. Autophagy receptors function in the selective autophagic clearance of disease-related proteins via ubiquitin signals. Thus, augmentation of autophagy is potentially a good therapeutic approach for neurodegenerative diseases. This review focuses on the role of ubiquitin signals in autophagy and their relevance to the onset of neurodegenerative diseases.

## 2. Neurodegenerative Diseases and Protein Aggregates

AD is a progressive neurodegenerative disorder that leads to cognitive decline [5]. The main hallmarks of AD are deposition of β-amyloid protein (Aβ) outside neurons, termed senile plaques, and in the vascular walls of the brain, and the accumulation of hyperphosphorylated Tau-protein as neurofibrillary tangles inside neurons [6]. Aβ is generated from amyloid precursor protein (APP) through sequential cleavage by β-secretase and γ-secretase complexes [7]. Studies of familial AD show that AD-causing variants in genes encoding APP and presenilins, catalytic components of the γ-secretase complex, elevate relative levels of the Aβ1–42 or Aβ1–43 isoforms of Aβ1–40 [8]. Aβ1–42 and Aβ1–43 are more aggregation-prone and cytotoxic compared with the Aβ1–40 peptide [9]. Initially, Aβ deposits are found exclusively in the neocortex and subsequently expand into the hippocampus, striatum, and brainstem [10]. Aβ is produced from APP and is mainly secreted into the extracellular space [11]. However, Aβ oligomers also accumulate intracellularly through endocytosis of secreted Aβ [12]. A triple-transgenic model of familial AD harboring transgenes expressing PS1(M146V), APP(Swe), and Tau(P301L) was defective for synaptic plasticity, including long-term potentiation, because of the accumulation of intraneuronal Aβ [13]. These findings provide evidence that extra- and intra-cellular Aβ accumulation causes cognitive impairment. 

Aβ accumulation is the initial event in AD progression. Subsequently, Tau pathology develops in a delayed fashion [14]. Tau promotes the assembly of tubulin into microtubules, a component of the cytoskeleton, and lends support to neuronal morphology [15]. In normal brain, Tau contains two phosphates per molecule, while in AD, fibrillary Tau is abnormally phosphorylated (approximately eight phosphates per molecule) [16]. This hyper-phosphorylation is mediated by GSK3β, and it affects the interaction of Tau with microtubules, leading to neurodegeneration and cognitive impairment [17,18]. Accumulation of abnormally phosphorylated Tau is also causative for other neurodegenerative disorders, including frontotemporal lobar degeneration (FTD), corticobasal degeneration, and progressive supranuclear palsy [19]. Intracellular Tau aggregates have been suggested to spread though synaptic circuits in a prion-like manner [20]. Neurofibrillary tangles, a major pathological hallmark of AD, are found at early stages in the transentorhinal cortex and the entorhinal cortex, a region providing input to the hippocampal circuitry [19]. Subsequently, Tau pathology propagates to the hippocampus, the temporal cortex, and then progresses to primary motor/sensory areas [19].

Parkinson’s disease and dementia with Lewy bodies are neurodegenerative diseases that are characterized by the presence of intracellular abnormal deposits called Lewy bodies and Lewy neurites. These deposits mainly consist of α-synuclein, which is a natively unfolded protein localized to the nucleus and presynaptic nerve terminals [21,22,23]. Several α-synuclein variants, such as missense and multiplication variants, are responsible for familial PD, suggesting that increased expression and abnormal structure of α-synuclein cause its aggregation and neurodegeneration [24,25]. Although accumulated α-synuclein is usually modified by phosphorylation, nitrosylation, glycation, and glycosylation, it remains unclear whether the aggregation is linked to its modification [26,27,28]. Fibrillar α-synuclein can propagate Lewy body and Lewy neurite pathology through cell-to-cell transmission leading to synaptic dysfunction and death of dopaminergic neurons in in vitro primary neurons and in the mouse brain [29,30,31]. Indeed, Lewy body-like inclusions were propagated in grafted embryonic nigral neurons that were transplanted into PD patients [32,33]. α-Synuclein is detected in cerebrospinal fluid of subjects with or without PD [34]. The mechanism by which α-synuclein is secreted remains unclear although the involvement of exosomal release and exocytosis via vesicles or multivesicular bodies has been reported [35,36,37]. Uptake of extracellular α-synuclein has been proposed to be mediated by pinocytosis and receptor-mediated endocytosis. Extracellular α-synuclein fibrils bind cell surface heparan sulfate proteoglycans and are intracellularly taken up through pinocytosis [38]. Tau fibrils are also incorporated into cells in the same manner [38]. In addition, lymphocyte-activation gene 3 (LAG3), a transmembrane protein, has high affinity for α-synuclein fibrils [39]. Interestingly, pathological Tau and Aβ species do not bind to LAG3, indicating that LAG3 is a specific receptor for α-synuclein fibrils [39]. LAG3 deficiency effectively reduces the endocytosis of α-synuclein fibrils and the propagation of PD pathology [39]. 

Amyotrophic lateral sclerosis (ALS) and FTD are neurodegenerative diseases characterized by motor and cognitive impairment, respectively. Both diseases have genetic and pathological overlaps. For example, variations in various genes, such as TDP-43, FUS, p62 and C9 or f72, are attributed to the etiology of both diseases, and TDP-43 pathology is often observed in both diseases (in ~97% of ALS and ~50% of FTD cases) [40,41]. In most cases of familial and sporadic ALS, immunohistochemical analysis shows that TDP-43 is included in ubiquitin-positive round and skein-like inclusions [42]. However, TDP-43 pathology is negative in other cases of familial ALS, such as in cases with SOD1 and FUS variations where inclusions are composed of SOD1 and FUS, respectively [43]. Numerous studies demonstrate that these mutant forms are aggregate-prone proteins that are cytotoxic by causing dysfunction to various cellular processes [44,45]. TDP-43, SOD1, and FUS aggregates also have prion-like seeding activity, which propagates ALS pathology [27,46,47,48]. TDP-43 and FUS are RNA-binding proteins that are involved in RNA and protein quality control [49]. Exposure to various stresses, such as heat shock, oxidative stress, and endoplasmic reticulum stress, induces formation of stress granules, which are dynamic assemblies of proteins and RNAs [50]. Stress granules are membrane-less organelles that contain translationally stalled mRNAs associated with translation initiation factors and multiple RNA-binding proteins, suggesting that stress granules regulate mRNA translation and stability and protect from environmental stresses [51,52]. Recent reports have indicated that variations linked to ALS-FTD in TDP-43, FUS, TIA-1, and C9 or f72 cause abnormal stress granule assembly and disassembly. For example, the TDP-43^A382T^ ALS-FTD variation causes a significant reduction in stress granule assembly in human fibroblasts [53]. In contrast, the TIA-1^P362L^ ALS-FTD variation delays stress granule disassembly and promotes the accumulation of non-dynamic stress granules [54]. These results indicate that dysfunctional stress granule dynamics might contribute to ALS pathogenesis. Moreover, mutations of p62 and OPTN were also identified in familial and sporadic ALS-FTD [55]. Both proteins are known as a ubiquitin binding protein shuttling ubiquitinated proteins for their degradation [55]. The FUS-containing inclusions are also immunoreactive with antibodies to p62 and OPTN in spinal anterior horn neurons in all sporadic ALS and in non-SOD1-familial ALS cases [55]. Recently, it was reported that ALS-FTLD-linked mutations of p62 disrupt autophagy and anti-oxidative stress pathway underlying the neurotoxicity in ALS-FTLD [56].

Other neurodegenerative diseases are also characterized by neuronal protein aggregates. Expanded polyglutamine (polyQ) tracts are aggregation-prone and expanded polyQ-containing proteins, such as huntingtin and ataxins, cause HD and spinocerebellar ataxia, respectively [57]. Huntingtin is a 348 kDa protein, and its N-terminal region contains the expandable polyQ tract [58,59]. Huntingtin undergoes post-translational modifications at multiple sites, such as phosphorylation, acetylation, sumoylation and ubiquitination, and is then cleaved by various proteases [58,60]. Cleaved N-terminal fragments with an expanded polyQ tract are released and form fibrillary aggregates or inclusion bodies [61]. Pathogenic polyQ-expanded huntingtin also has prion-like properties. Mutant huntingtin aggregates were detected in the extracellular matrix of grafted neurons in HD patient brains, indicating that pathological huntingtin can spread within the brain [62,63]. 

## 3. Ubiquitination in Protein Degradation

Accumulation of harmful proteins is a hallmark of various neurodegenerative diseases, as described above. Cells are protected from harmful proteins by protein quality control mechanisms, including molecular chaperone and protein degradation systems [64]. Eukaryotes have two major protein degradation systems, the ubiquitin–proteasome system (UPS) and the autophagy–lysosome pathway (ALP). In the UPS, ubiquitin-tagged proteins are targeted by a multi-subunit protease complex, the proteasome. The proteasome consists of a multi-catalytic proteinase complex (20S) and two regulatory complexes (19S and 11S) [65]. Misfolded proteins and short-lived proteins undergo ubiquitination by a multi-step process that requires a ubiquitin-activating (E1) enzyme, a ubiquitin-conjugating (E2) enzyme, and a ubiquitin ligase (E3) [66]. Initially, ubiquitin is activated by the E1 enzyme in an ATP-dependent manner, and activated ubiquitin is then transferred to an E2 enzyme. E3 then ligates the ubiquitin to the target protein [3]. Ubiquitinated proteins are recruited to the regulatory complex of the proteasome and are then deubiquitinated and unfolded [67]. Linearized proteins then translocate into the proteolytic chamber of the 20S proteasome and are cleaved by its six proteolytic sites [68]. 

The ALP is an intracellular metabolic process in which cytoplasmic proteins and organelles sequestrated by autophagosomes are degraded in lysosomes [69]. Autophagy is regulated by more than 30 autophagy regulated proteins and its core machinery is classified in four subgroups: (1) The ATG1/ULK1 complex; (2) ATG9 and its cycling system; (3) the phosphatidylinositol 3-kinase complex; (4) two ubiquitin-like conjugation systems (ATG8/LC3 and ATG12) [69]. In mammals, the ULK1 complex has an essential role in the initiation of autophagy and is directed to the endoplasmic reticulum together with ATG9 vesicles and the phosphatidylinositol 3-kinase complex [69]. DFCP1 and WIPIs are recruited to the endoplasmic reticulum membrane and promote the formation of isolation membrane [69]. Autophagosome formation is mediated by two ubiquitin-like conjugation systems, conjugation of ATG12 to ATG5 and conversion of LC3 to a phosphatidylethanolamine-conjugated membrane-bound form [69]. Finally, mature autophagosomes fuse with lysosomes, resulting in degradation of cellular components [69].

Mono-ubiquitination functions in the regulation of protein interaction, trafficking, and transcriptional activity [70] and mono-ubiquitinated proteins are degraded by the proteasome in both mammalian and yeast cells [71]. Multi-ubiquitin is generated by the sequential conjugation of ubiquitin to ubiquitin via Lys residues (Lys-6, Lys-11, Lys-27, Lys-29, Lys-33, Lys-48, and Lys-63) or Met residues (Met-1, Met-14, Met-20) [3]. Lys-11-, Lys-48-, and Lys-63-linked poly-ubiquitination act as proteolytic signals for the proteasome and autophagy (Figure 1). Lys-48-linked poly-ubiquitin is the major signal of numerous short-lived proteins and unfolded proteins for proteasomal degradation [72]. Lys-11-linked poly-ubiquitin is also involved in degradation of short-lived cell cycle proteins and in the ERAD (endoplasmic reticulum-associated degradation) pathway upon ER stress [73]. Furthermore, small aggregated proteins are selectively degraded via the autophagy–lysosome system. Lys-48- and Lys-63-linked poly-ubiquitin is required for selective sequestration of aggregated proteins into autophagosomes through autophagy receptor proteins [3]. Furthermore, accumulation of Lys-11-, Lys-48-, Lys-63-linked poly-ubiquitinated insoluble proteins was observed in the brains of *Atg5*- and *Atg7*-null mice, indicating that multiple ubiquitin signals might be involved in autophagic degradation of various cargoes (Figure 1) [74]. Autophagy receptor proteins possess both a ubiquitin-binding domain and an LC3-interacting region (LIR), and bind to various cargoes, such as protein aggregates, intracellular organelles and microbial invaders [75,76]. Ubiquitinated protein aggregates are selectively recognized by autophagy receptor p62, NBR1, OPTN, and TOLLIP [77,78]. These complexes are then associated with autophagosome protein LC3 though an LIR domain, resulting in sequestration into an isolation membrane and degradation in lysosomes [78]. Ubiquitination is also often required for selective clearance of organelles. For example, damaged mitochondria in PD are eliminated via ubiquitin-dependent PINK1-Parkin-mediated mitophagy [79]. Upon mitochondrial damage, Parkin, an E3 ubiquitin ligase, conjugates Lys-6-, Lys-11-, Lys-48-, and Lys-63-linked poly-ubiquitin to mitochondrial outer membrane proteins, and then mitochondria bind to autophagy receptors [79,80,81]. Furthermore, PINK1 coordinately acts upstream of Parkin in this process. Phosphorylated ubiquitin by PINK1 is required for Parkin activation [82]. These findings indicate that ubiquitin signals also have an important role in the autophagic clearance of organelles.

## 4. Ubiquitination of Neurodegenerative Disease-Associated Proteins

Harmful proteins causing neurodegenerative diseases undergo ubiquitination and pathological analyses using anti-ubiquitin antibodies identify various protein aggregates and inclusions. Paired helical filament-Tau (PHF-Tau) is modified by Lys-6-, Lys-11-, Lys-48-, Lys-63-poly-ubiquitin chains and mono-ubiquitin in AD brains or cultured cells (Figure 2) [83,84,85]. CHIP, a HSP70 co-chaperone, is an E3 ubiquitin ligase of PHF-Tau [84,86]. Lys-254, 257, 311, and 317 of PHF-Tau are acceptors for ubiquitin [85]. Lys-724, 725, 726, 751, and 763 of APP is intracellularly conjugated with ubiquitin in mouse brain [87] and impairment of this ubiquitination leads to accumulation of both secreted and intracellular Aβ40 [87]. 

α-Synuclein undergoes ubiquitination by various E3 ubiquitin ligases (Figure 2). Seven in absentia homolog (SIAH), an E3 ubiquitin ligase, mono-ubiquitinates α-synuclein at Lys-12, 21, and 23, resulting in an increase in the aggregation of α-synuclein and apoptotic cell death [88,89]. NEDD4 ubiquitin ligase also targets α-synuclein and mediates Lys-63-poly-ubiquitin [90]. Ubiquitinated α-synuclein is degraded by the endosomal-lysosomal pathway, suggesting that this process might have a protective effect against the pathogenesis of PD and other α-synucleinopathies [90]. Furthermore, CHIP is also involved in α-synuclein mono-ubiquitination or poly-ubiquitination, similarly of PHF-Tau [91,92]. CHIP-mono-ubiquitinated α-synuclein is deubiquitinated by USP9X. USP9X knockdown promotes accumulation of mono-ubiquitinated α-synuclein and enhances the formation of α-synuclein inclusions upon proteolytic inhibition [92]. Ubiquitin ligase E6-AP is localized to Lewy bodies in the PD brain, and is involved in α-synuclein ubiquitination and proteasome-dependent degradation [93]. 

ALS-causing TDP-43 and SOD1 aggregates are also detected by anti-ubiquitin antibodies. TDP-43 is targeted by Znf179 ubiquitin ligase and is modified by poly-ubiquitin chains [94]. Znf179 knockout suppresses TDP-43 proteosomal turnover, resulting in accumulation of insoluble TDP-43 and cytosolic TDP-43 inclusions in the cortex, hippocampus and midbrain regions [94]. In addition to Znf179, CUL2 ubiquitin ligase can modify misfolded TDP-43 with poly-ubiquitin, coordinately with von Hippel Lindau protein (VHL) [95]. Mass spectrometry analysis identified TDP-43 ubiquitination sites to be Lys-84, Lys-95 Lys-160, Lys-181, and Lys-263 residues and that its poly-ubiquitin chains link via Lys-48 and Lys-63 [96]. SOD1 is targeted by NEDL1 and gp78 ubiquitin ligases. NEDL1 colocalizes with SOD1 inclusions in the spinal cord ventral horn motor neurons of both ALS patients and mutant SOD1 transgenic mice [97]. gp78 ubiquitin ligase is also involved in the ubiquitination of SOD1 [83]. gp78 is a protein with at least five membrane-spanning domains, including a RING finger consensus sequence, and plays an important role in ERAD [98]. Interestingly, this ubiquitin ligase also mediates ubiquitination of spinocerebellar ataxia-associated ataxin-3 [99]. gp78 overexpression promotes the ubiquitination and degradation of SOD1 and ataxin-3 in cultured cells, whereas knockdown of gp78 stabilizes them [99].

Turnover of huntingtin is regulated by ubiquitination, via Lys-48- and Lys-63-poly-ubiquitin. Although aggregated mutant huntingtin mainly includes Lys-63-poly-ubiquitin chains, overexpression of Lys-48-specific ubiquitin ligase, Ube3a, reduces Lys-63-ubiquitination and huntingtin aggregation, enhancing its degradation via the Lys-48 ubiquitin–proteasome system [100]. Similarly, ubiquitin ligase, UBR5, is also involved in Lys-48-proteasomal degradation of both normal and mutant huntingtin [101]. However, tumor necrosis factor receptor-associated factor 6 (TRAF6) can promote the Lys-63-ubiquitin chain on mutant huntingtin and might contribute to autophagic clearance of huntingtin aggregates [102]. Collectively, numerous neurodegenerative disease-associated proteins undergo ubiquitination by a variety of ubiquitin ligases. These ubiquitin signals mainly serve to eliminate pathogenic proteins, although the ubiquitin signal on neurodegenerative disease-associated proteins can be pathogenic.

## 5. Autophagic Degradation of Neurodegenerative Disease-Associated Proteins

Small protein aggregates are thought to be degraded by ALP. Autophagy receptors recognize ubiquitin chains bound to cargoes and transport them to autophagosomes (Figure 3A). Recent advances in mass spectrometry technology have contributed to the decoding of ubiquitin signals and have revealed the diversity of ubiquitin chains. In addition, the ubiquitin binding-domains of autophagy receptors have been categorized (Figure 3B). p62, Nbr1, and c-Cbl have a UBA domain, a small domain of about 40 residues [103]. The UBA domain of p62 and Nbr1 binds strongly to both Lys-48- and Lys-63-poly-ubiquitin [104]. Moreover, the UBA domain of ubiquilin-1 and yeast Ede1, other ubiquitin binding proteins, have a high affinity for mono-ubiquitin, indicating that autophagy receptors with the UBA domain might bind to mono-ubiquitin [105,106]. OPTN has two ubiquitin binding domains, a UBAN and a zinc finger domain. A UBAN domain can interact not only with Lys-63-linked poly-ubiquitin but also with linear ubiquitin chains, which are generated between the N-terminal methionine of one ubiquitin and the C-terminal glycine of the next in the chain. (Figure 3B) [107,108]. However, the zinc finger domain of OPTN recognizes various protein aggregates in a ubiquitin-independent manner [109], although the same domain of NDP52 can recognize mono-ubiquitin, Lys-48-, and Lys-63-poly-ubiquitin [110]. NDP52 is a selective autophagy receptor for cytosolic bacteria (xenophagy) and damaged mitochondria (mitophagy), which are decorated with ubiquitin [111,112]. Although preferences of ubiquitin binding domains for ubiquitin codes remain unclear, individual autophagy receptors might selectively recognize disease-associated aggregates by the ubiquitin code. Indeed, the UBA domain of p62 can bind both Lys-48-linked and Lys-63-linked ubiquitin chains but has a higher affinity for Lys-63 chains [113]. In addition, the UBA domain of NBR1 is structurally distinct from the p62 UBA domain, resulting in a different interaction with ubiquitin. NBR1 has significantly higher affinity for mono-ubiquitin compared with p62 [114]. Accordingly, substrate preference of autophagy receptors might be dependent on the ubiquitin codes on cargoes (Figure 3A).

Activation of autophagy receptors is mediated by various kinases. For example, p62 is phosphorylated by various kinases, such as mTORC1, casein kinase 1, and TBK1 [64,115,116]. OPTN and NBR1 activities are also regulated by TBK1 and GSK3β, respectively [117,118]. Several phosphorylation sites are located in the ubiquitin binding domain, whose status alters the affinity for ubiquitinated proteins. Indeed, inhibition of several phosphorylation sites reduces cargo-binding potential [119]. This evidence indicates that structural alterations to autophagy receptors by phosphorylation controls autophagic clearance of various cargoes. 

Autophagy activation accelerates elimination of neurodegenerative disease-associated protein aggregates and inclusions. The mTORC1 inhibitor, rapamycin, is well known to induce autophagy activity [120]. The effect of rapamycin has been investigated using various neurodegenerative disease models. For example, accumulation of Tau, huntingtin, and α-synuclein aggregates was significantly decreased in cultured cell and Drosophila models of AD, HD, and PD [121,122]. Moreover, the mTORC1-independent autophagy inducer, trehalose, also reduced protein aggregation and neuronal degeneration in ALS and tauopathy model mice [123,124]. Moreover, progression of PD-like pathology was investigated in autophagy suppressor Rubicon-KO mice, in which basal autophagy is constitutively activated. Spread of Lewy body-like α-synuclein aggregates was significantly reduced in the brain of this mouse [125]. These results indicate that autophagy induction may be an effective treatment for various neurodegenerative diseases.

## 6. Concluding Remarks

Protein quality control systems, such as UPS and ALP, decline with age, which is a leading cause of neurodegenerative diseases. Clinical trials of several autophagy activators have been conducted for AD and ALS patients. For example, resveratrol is a natural polyphenol that induces autophagy activity by directly inhibiting mTOR [126]. In individuals with mild to moderate AD, decline of cerebrospinal fluid and plasma Aβ1–40 levels were observed in a resveratrol-treated group compared with a placebo-treated group [127]. However, improvement of cognitive function was not reported in this trial. The existing drugs, metformin and lithium, are also autophagy inducers, and clinical trials of these drugs for AD patients have also been conducted [128,129,130]. Currently, clinical trials of rapamycin are planned for ALS patients but not for AD patients, although it is hoped that further evidence warranting trials in AD patients will be forthcoming. While, it has been revealed that deubiquitinating enzymes such as UCL-L1 and ubiquitin-specific proteases are also involved in PD and AD through proteostasis [131]. Various inhibitors of deubiquitinating enzymes might be a new therapeutic target [131]. Collectively, detailed knowledge of ubiquitin chains in neurodegenerative disease-associated proteins and structural analyses of their interactions with ubiquitin binding domains will be beneficial for the development of novel therapies for neurodegenerative diseases.

## Figures and Tables

**Figure 1 cells-09-02022-f001:**
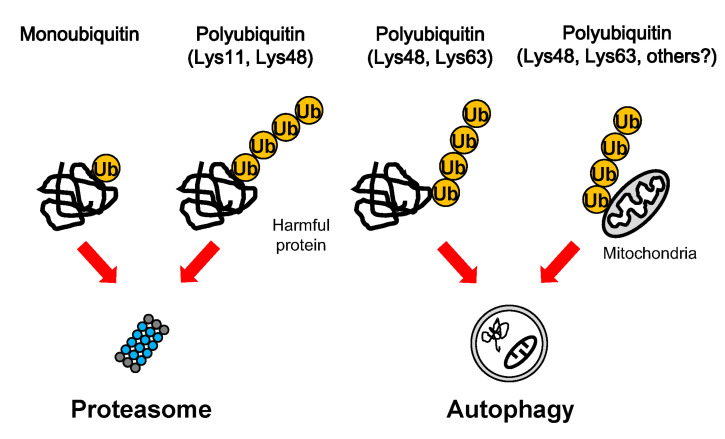
Degradation of ubiquitinated cargoes. Harmful proteins and mitochondria can be modified by various ubiquitin additions, such as mono-ubiquitin and Lys-11-, Lys-48-, and Lys-63-poly-ubiquitin chains. The proteasome preferentially degrades mono-ubiquitinated proteins and Lys11- and Lys-48-linked proteins, whereas autophagy preferentially eliminates Lys-48-, and Lys-63- decorated protein aggregates and mitochondria.

**Figure 2 cells-09-02022-f002:**
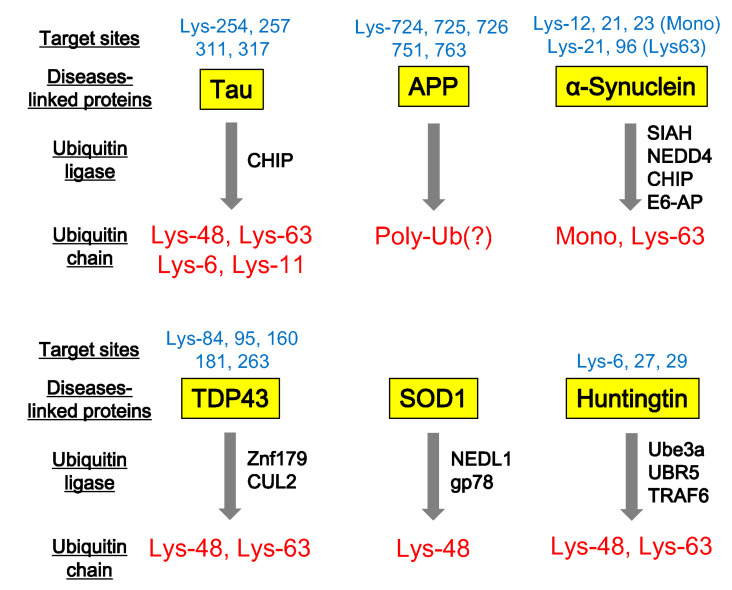
Ubiquitination of neurodegenerative disease-associated proteins. Neurodegenerative disease-associated proteins, such as Tau, APP (β-amyloid precursor protein), α-synuclein, TDP43, SOD1, and Huntingtin are ubiquitinated at individual target sites. Specific ubiquitin ligases involved in this ubiquitination and the pattern of ubiquitin chains can be identified by various biochemical studies.

**Figure 3 cells-09-02022-f003:**
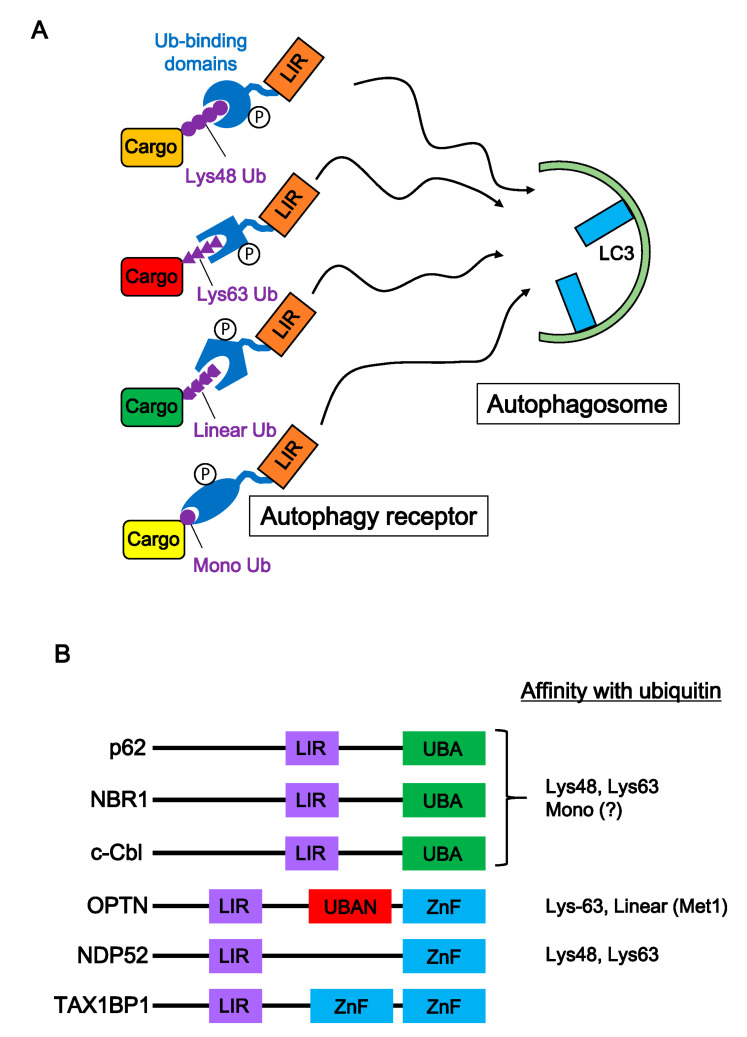
Autophagy receptors and selective autophagy of ubiquitinated cargoes. (**A**) Autophagy receptors play an important role in selective autophagy. Appropriate autophagy receptors bind to various ubiquitinated cargoes (Lys-48-, Lys-63-, linear-poly-ubiquitin chain, and mono-ubiquitin) through their ubiquitin binding domain. Many autophagy receptors are regulated by various kinases, and then autophagy receptor-cargo complexes interact with the autophagosome protein, LC3. (**B**) Structure of major autophagy receptors. Autophagy receptors consist of an LIR domain and a ubiquitin binding domain. Ubiquitin binding domains (UBA, UBAN, and ZnF) are classified based on motif sequences. Biochemical studies show preferences of autophagy receptors for ubiquitin types.

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
