# Peer review of "Ubiquitin, Autophagy and Neurodegenerative Diseases"

_cells, 2020, doi:10.3390/cells9092022_

Round 1
Reviewer 1 Report
[Cells] Manuscript ID: cells-906658
Ubiquitin, Autophagy and Neurodegenerative Diseases
RECOMMENDATION - MINOR REVISION
Overall this is a good, useful and thorough review describing the ubiquitination in the process of autophagy and neurodegenerations.
Comment
- The Authors should provide more information about novel therapies based on deubiquitinating enzymes. Maybe limitations of this kind of therapy?
e.g.
Lim KH, Joo JY, Baek KH. The potential roles of deubiquitinating enzymes in brain diseases. Ageing Res Rev. 2020;61:101088. doi:10.1016/j.arr.2020.101088
Author Response
Thank you for the thoughtful and constructive feedback you provided regarding our manuscript. We have added the parts you pointed out in the Concluding remarks section as described below.
While, it has been revealed that deubiquitinating enzymes such as UCL-L1 and ubiquitin-specific proteases are also involved in PD and AD through proteostasis [128]. Various inhibitors of deubiquitinating enzymes might be a new therapeutic target [128].
Reviewer 2 Report
In this review, the authors went through more than 120 papers and summarized current knowledge regarding the roles of ubiquitin-proteasome system (UPS) and the autophay lysosome pathway in the context of some neurodegenerative disorders. Although the spectrum of brain disorders in which both systems are affected, the authors were able to summarize the most important aspects of the most common diseases. It was a very nice piece of work, very confortable and pleasant to read. English language is fine.
Neverthless, there are some information that this reviewer would like to see as well in the manuscript, and some minor changes.
- Line33- it is mentioned that homotypic polyub chains are generated by conjugation of two ub molecules.. However, the sentence is not fully correct. Homotypic chains are formed of two or more ubiquitin proteins within the same internal lysine residue. Therefore, the current sentence might be misleading.
- The protein p62/SQSTM1 is also a major factor in familial and sporadic forms of ALS/FTD. The reviewer suggests that this protein should be discussed as well in section ´2. Neurodegenerative diseases and proteins aggregates` (e.g PNAS. 2017;114(39):E8294-E8303).
- In section `3. Protein ubiquitination in protein degradation`, it would be important to discuss the potential role of Ubiquitin phosphorylation by PINK1 in regulating the mitophagy process in Parkinson´s Disease (e.g Nature. 2014 Jun 5;510(7503):162-6)
- It would also be important if a small section discussing the role of other forms of autophagy, in a neurodegenerative context is added to the manuscript.
Author Response
Thank you for the thoughtful and constructive feedback you provided regarding our manuscript. In accordance with the reviewer's comment, we have added the relationship of p62 and ALS and PINK1-mediated ubiquitin phosphorylation as described below.
Line117-123
Moreover, mutations of p62 and OPTN were also identified in familial and sporadic ALS-FTD [55]. Both proteins are known as a ubiquitin binding protein shuttling ubiquitinated proteins for their degradation [55]. The FUS-containing inclusions are also immunoreactive with antibodies to p62 and OPTN in spinal anterior horn neurons in all sporadic ALS and in non-SOD1-familial ALS cases [55]. Recently, it was reported that ALS-FTLD-linked mutations of p62 disrupt autophagy and anti-oxidative stress pathway underlying the neurotoxicity in ALS-FTLD [56].
Line185-186
Furthermore, PINK1 coordinately acts upstream of Parkin in this process. Phosphorylated ubiquitin by PINK1 is required for Parkin activation [82].
We have corrected Line33, as described below.
Homotypic poly-ubiquitin chains are generated by conjugation of two or more ubiquitin molecules via their seven lysine residues...
Reviewer pointed out to discuss other autophagy systems in neurodegenerative disorders. Chaperone-mediated autophagy is known to be involved in PD. But, ubiquitin dose not participate in this system. Accordingly, we we have not added this in our manuscript.
Reviewer 3 Report
In the review article, entitled "Ubiquitin, Autophagy, and Neurodegenerative Diseases" authors made an attempt to review all relevant literature regarding ubiquitination, protein degradation, autophagy in relation to neurodegenerative diseases. The article is well written. But I recommend moderate English language corrections to improve the readability. Especially the introduction could have been better.
Author Response
Thank you for the thoughtful and constructive feedback you provided regarding our manuscript. In accordance with the reviewer's comment, we have revised our manuscript. The writing in red is the corrected places. I had a native person (Jeremy Allen, PhD) check the manuscript.
Reviewer 4 Report
The paper: Ubiquitin, Autophagy and Neurodegenerative Diseases , submitted by:
Yoshihisa Watanabe1,*, Katsutoshi Taguchi2 and Masaki Tanaka2,*
is a very well written and comprehensive review covering three large areas, the ubiquitin proteasome system, neurodegenerative diseases and autophagy. The authors managed to compress a lot of scientific data in a relatively short paper, without losing information or overview. The diseases: Alzheimer's disease (AD), Parkinson's disease (PD), amyotrophic lateral sclerosis (ALS), and Huntington’s disease are introduced in a way to find a good link to autophagy and the ubiquitin proteasome system, which are sufficiently introduced as well. All figures are very good, concise and self explaining. All major publications can be found in the used literature.
It was a joy to read.
Author Response
Thank you for taking the time and the good rating.